# Effectiveness of Moderate-Intensity Aerobic Water Exercise during Pregnancy on Quality of Life and Postpartum Depression: A Multi-Center, Randomized Controlled Trial

**DOI:** 10.3390/jcm10112432

**Published:** 2021-05-30

**Authors:** Araceli Navas, María del Carmen Carrascosa, Catalina Artigues, Silvia Ortas, Elena Portells, Aina Soler, Aina M. Yañez, Miquel Bennasar-Veny, Alfonso Leiva

**Affiliations:** 1Hospital Comarcal de Inca, Balearic Islands Health Services, 07300 Inca, Spain; araceli.navas@hcin.es; 2Mallorca Primary Health Care, Balearic Islands Health Services, 07002 Palma, Spain; mccarrascosa@ibsalut.caib.es (M.d.C.C.); cartigues@ibsalut.caib.es (C.A.); silviaortas@hotmail.com (S.O.); meportells@ibsalut.caib.es (E.P.); 3Primary Care Research Unit of Mallorca, Balearic Islands Health Services, 07002 Palma, Spain; asoler@ibsalut.caib.es (A.S.); aleiva@ibsalut.caib.es (A.L.); 4Health Research Institute of the Balearic Islands (IdISBa), 07120 Palma, Spain; 5Nursing and Physiotherapy Department, Balearic Islands University, 07122 Palma, Spain

**Keywords:** exercise, delivery, postpartum depression, epidural anesthesia, caesarian, natural childbirth

## Abstract

Background: The global prevalence of postpartum depression is about 20%. This disease has serious consequences for women, their infants, and their families. The aim of this randomized clinical trial was to analyze the effectiveness and safety of a moderate-intensity aerobic water exercise program on postpartum depression, sleep problems, and quality of life in women at one month after delivery. Methods: This was a multi-center, parallel, randomized, evaluator blinded, controlled trial in a primary care setting. Pregnant women (14–20 weeks gestational age) who had low risk of complications and were from five primary care centers in the area covered by the obstetrics unit of Son Llatzer Hospital (Mallorca, Spain) were invited to participate. A total of 320 pregnant women were randomly assigned to two groups, an intervention group (moderate aquatic aerobic exercise) and a control group (usual prenatal care). One month after birth, sleep quality (MOS sleep), quality of life (EQ-5D), and presence of anxiety or depression (EPDS) were recorded. Findings: Women in the intervention group were less likely to report anxiety or depression on the EQ5D (11.5% vs. 22.7%; *p* < 0.05) and had a lower mean EPDS score (6.1 ± 1.9 vs. 6.8 ± 2.4, *p* < 0.010). The two groups had no significant differences in other outcomes, maternal adverse events, and indicators of the newborn status. Conclusion: Moderate-intensity aquatic exercise during pregnancy decreased postpartum anxiety and depressive symptoms in mothers and was safe for mothers and their newborns.

## 1. Introduction

Depression is a leading cause of disability worldwide and is a major contributor to the overall global disease burden [1]. Worldwide, more than 264 million people of all ages suffer from depression [2], and depression is two times more common in women during childbearing years than in men [3]. Perinatal depression, defined as major or minor depressive episodes during pregnancy for the first 12 months after delivery, is one of the most common medical complications during pregnancy and the postpartum period [4].

Previous studies of women during pregnancy and after childbirth reported that the prevalence of perinatal depression ranged from 10 to 20% [4,5,6,7]. The largest meta-analysis and meta-regression to date of postpartum depression (PPD) in women after childbirth reported that the global prevalence of PPD was approximately 17.7% [8]. This prevalence is higher among women from low- and low-middle-income countries, women with poor social support, and women with low socioeconomic status, and PPD thus contributes to worldwide health inequalities [9]. In Spain, the prevalence of prenatal depression is about 14.8% [10] and the prevalence of PPD is 10–13%, although there have only been limited studies of this topic in Spain [11,12,13].

PPD has adverse effects on women, their infants, and their families. In particular, PPD can adversely affect the mother–infant relationship and the child’s growth and development, including cognitive and language development. It also affects children by increasing the risk of behavioral problems, insecure or disorganized personal attachments, leaving school at a younger age, and depression at age 16 to 18 years [14,15]. PPD can also increase difficulties with breastfeeding, decrease the level of breastfeeding self-efficacy, and increase early cessation of breast feeding [3,16]. Suicide is a leading cause of death during the perinatal period in high-income countries, and it accounts for 5–20% of all maternal deaths [8,17,18].

Although PPD is a major health issue for many women from diverse cultures, it often remains undiagnosed [5,19,20]. This may be because clinicians attribute changes in a woman’s sleep, appetite, and libido to the normal changes that occur during pregnancy and the postpartum period [4].

Exercise can improve overall psychological well-being, and mental health professionals commonly recommend exercise for treatment of depression [21]. Several meta-analyses reported that exercise was associated with relief from depression [22,23]. Although the underlying mechanism is uncertain, there is evidence that exercise reduces the symptoms of depression during pregnancy and the postpartum period by increasing the levels of dopamine, serotonin, and noradrenalin [24,25].

Aquatic-aerobic exercise has some advantages including lower probability of miscarriage, less edema, increased diuresis, reduced arterial pressure, and less back pain [26] and also allows for emotional interactions among pregnant women [27].

There is a general agreement that exercise should be encouraged for all women during pregnancy (except those with contraindications for exercise) to reduce pregnancy complications [24,28]. However, the effects of exercise on a woman’s mental health during the postpartum period are uncertain. Recent systematic reviews and meta-analyses showed that physical activity prevented and effectively treated antenatal depression [29,30]. Other studies showed that initiation of aerobic exercise during the postpartum period reduced the symptoms of mild depression [31,32]. However, some studies reported contradictory findings regarding the effect of antenatal exercise on PPD [30,33,34,35,36,37,38,39,40]. These contradictory results could be explained by the type, intensity, frequency, and duration of the physical activity. Moreover, it could be due to the supervision of the exercise or the combination of physical activity with other interventions (diet, education, smoking) [37,38,39,40].

Women in the postpartum period experience high levels of sleep disturbances [41,42]. Previous studies have shown that sleep problems adversely affect mood [43] and reduce quality of life in mothers [44]. Sleep deprivation is also associated with antenatal stress and PPD [45,46]. Women in the postpartum period experience high levels of sleep disturbances due to irregular newborn sleep patterns and the demands of the newborn [41,42]. The association between sleep and depression may result from similar mechanisms [47]. Some studies have shown that exercise, especially water exercise, can improve the sleep quality of pregnant women [48,49].

In the present study, we analyzed the effectiveness and safety of a moderate aerobic water exercise program on the risk of PPD, sleep problems, and quality of life in women at 1 month after delivery. We hypothesized that women in moderate aerobic water exercise are at less risk of postnatal depression than women in the control group.

## 2. Materials and Methods

### 2.1. Study Design

This was a multi-center, parallel, randomized, evaluator-blinded, controlled clinical trial of pregnant women that was conducted from November 2014 to March 2017. A detailed description of the study protocol was published previously [50].

### 2.2. Participants

Pregnant women from 5 primary care centers in Mallorca (Spain) in the area covered by the obstetric unit of Son Llatzer Hospital were invited to participate.

The inclusion criteria were pregnancy, age of 18 to 40 years, gestational age of 14 to 20 weeks, singleton pregnancy, and low obstetric risk (low-risk women with uncomplicated singleton term pregnancies with a vertex presentation who are expected to have an uncomplicated birth) [51].

The exclusion criteria were complicated obstetric history with a stillbirth or neonatal death, ≥6 previous pregnancies, ≥3 consecutive miscarriages, previous fetal death in utero, previous mid-trimester loss/cervical incompetence/known uterine anomaly, previous early onset of pre-eclampsia (<32 weeks gestation), rhesus iso-immunization, complications during the current pregnancy (such as fetal abnormality), cardiac disease, essential hypertension, renal disease, pre-existing diabetes, severe anemia, severe asthma, severe psychiatric disorder, substance abuse or smoking more than 20 cigarettes/day, obesity (BMI > 35 kg/m^2^) or significant underweight (BMI < 17 kg/m^2^), recurrent urinary tract or vaginal infection, and inability to swim.

### 2.3. Sample Size

For the primary hypothesis, 320 women provided 80% power at 5% significance two-tailed alpha to detect differences in the primary outcome. For the secondary hypothesis, we had 80% power to detect a between-arm effect size of 0.22 (standardized difference) in the Depression Edinburgh questionnaire and MOS-sleep questionnaire and a reduction of 5% in the risk of postpartum depression.

### 2.4. Randomization and Blinding

A midwife from the research team invited eligible pregnant women to participate in the study during their visits at the primary care centers. Women who were willing to participate and eligible were enrolled after they read and signed an informed consent agreement. The women were randomly allocated (1:1) using a computer-generated randomization list in blocks of 6 to receive moderate aquatic-aerobic exercise with usual antenatal care (intervention group) or usual antenatal care alone (control group). The allocation was concealed using central randomization. The midwife contacted the research unit of the Mallorca primary care unit by telephone and was informed of the allocation of each woman.

### 2.5. Intervention

Women in the intervention group participated in 45 min of water aerobics classes three times weekly in an indoor pool (28–30 °C) for 5 months. The exercise intensity was designed so that each woman maintained an estimated heart rate of 55–65% of the maximum (140 bpm, based on recommendations of the American College of Sports Medicine [24,52]).

The aquatic aerobic exercise program was carried out exclusively by midwives and consisted of: (i) warm-up out of the water (5–7 min; stretching and warming up the neck, pectoral muscles, shoulders, back, quadriceps, and calves, and mobility training of pelvic girdle, feet, ankles, and knees); (ii) warm-up in the water (5–10 min; walking in the water, taking big steps and then small jumps, walking sideways, and walking forward and backwards); (iii) moderate aquatic exercise (20 min; 4 different sets of exercises and compound exercises that included coordinated breathing, with each set consisting of exercises of the arms, legs, lower back, and pelvic floor); (iv) breathing and relaxation exercises (5 min); and (v) playful exercises (5 min).

Women were asked to discontinue the intervention if any of the following events occurred: vaginal bleeding, premature rupture of membranes, intrauterine growth retardation, placenta previa, severe anemia, regular painful contractions, amniotic fluid leakage, dizziness, dyspnea before exertion, headache, chest pain, calf pain or swelling, muscle weakness affecting balance, preterm labor, decreased fetal movement, or any contraindications to being physically active [24].

Women in the usual care group only received standard antenatal care, but this could include advice regarding physical activity.

### 2.6. Data Collection

Sociodemographic and clinical data at baseline were collected using a questionnaire and patients’ medical records. The clinical history of the mother included obstetric characteristics, smoking status, social class (using the classification of the Spanish Society of Epidemiology [53] as non-manual workers for class I, II, and III and manual workers for class IV and V), educational level, physical activity (using the Spanish version of International Physical Activity Questionnaire-Short Form, IPAQ-SF) [54,55], sleep quality (using the MOS sleep scale) [56], and quality of life (using EuroQol Five Dimension questionnaire, EQ-5D) [57].

One month after birth, sleep quality (MOS sleep), quality of life (EQ-5D), and the presence of depression symptoms (using the Edinburgh Postnatal Depression Scale, EPDS) [58] were recorded during a follow-up visit. The EPDS results were categorized risk of depression (>10) or no risk of depression (≤10).

Women were asked to report any adverse events that could be related to physical activity at each medical visit and exercise class. They were also asked to consider adverse events, such as back pain, urinary tract infection, bleeding, and uterus contractions.

### 2.7. Statistical Analysis

All statistical analyses were performed using IBM SPSS Statistics version 23 (SPSS/IBM, Chicago, IL, USA) according to a predefined analysis plan [50]. The significance of differences in baseline characteristics of the control and intervention groups were determined. An intention to treat (ITT) analysis was used to analyze all clinical outcomes, and all results were reported in accordance with the Consolidated Standards of Reporting Trials (CONSORT) guidelines extension for cluster trials [59]. The effectiveness of the intervention on postnatal depression was assessed using ANOVA tests for continuous measures and a logistic regression model for the categorical measures. Sensitivity analysis included the effectiveness of the intervention adjusted for unbalanced variables at baseline. All estimates included 95% confidence intervals (CIs), and all treatment effects were considered significant if the two-sided *p*-value was below 0.05. Multiple imputation was used for the main analysis because it generally provides less biased estimates of an effect than a complete cases analysis. Missing outcomes were accounted for using multiple imputation with chained equation [60].

### 2.8. Ethical Considerations

This study followed the principles outlined in the Declaration of Helsinki [61]. The study protocol was approved by the Primary Care Research Committee and the Balearic Ethical Committee of Clinical Research (registered CEI-IB Ref. No: 2358/14). All participants provided written informed consent, and were told that participation was voluntary and that they could withdraw at any time without any negative consequences on their medical treatments. Women at risk of depression (EPDS > 10) were referred to primary health care service by the research team.

The study was registered with ISRCTN (https://www.isrctn.com/ISRCTN14097513) on 4 September 2017. The results are presented clearly, honestly, and without fabrication, falsification, or inappropriate data manipulation. There are no restrictions on publicly sharing the dataset. However, because all participants provided informed consent, an anonymized minimal dataset is available (https://doi.org/10.5281/zenodo.3580637, accessed on 27 January 2021).

## 3. Results

We initially invited 380 women to participate (Figure 1). However, 5 women did not meet the inclusion/exclusion criteria, 10 could not be contacted to sign the informed consent document, and 71 refused to participate. Thus, we randomized the remaining 294 women to the exercise group (*n* = 148) or the usual care group (*n* = 146). There were two severe adverse events, and 21 women were lost during follow-up. At 1 month after birth, we had complete data about quality of life, depression, and sleep quality from 139 women in the intervention group and 132 women in the control group.

The baseline clinical and demographic characteristics of the two groups were similar (Table 1). However more women in the intervention group were smokers and physically inactive, and fewer women in the intervention group were in social class IV/V.

Analysis of maternal outcomes (Table 2), including depression, quality of life, and quality of sleep, indicated that significantly fewer women in the intervention group had anxiety or depression symptoms based on the EQ5D questionnaire (11.5% vs. 22.7%, *p* = 0.020).

In addition, the mean EPDS score was significantly lower in the exercise group (6.1 ± 1.9 vs. 6.8 ± 2.4, *p* = 0.010) and the risk of depression (EDPS > 10) was marginally lower in the exercise group (1.5% vs. 6.1%, *p* = 0.052).

The global Sleep Problems Index and almost all specific sleep problems evaluated by the MOS sleep questionnaire (except for awakening with short breath or headache) were less prevalent in the exercise group, although these differences were not statistically significant.

Analysis of adverse events and newborn outcomes (Table 3) indicated that the two groups had no significant differences in maternal adverse events (including urinary infection, bleeding, and back pain) or newborn status (Apgar score, pH of umbilical cord, intrapartum fetal distress). There was one fetal death in the exercise group and one abortion in the control group. In addition, 1 mother from the intervention group, 10 newborns from the intervention group, and 5 newborns from the control group were admitted to an intensive care unit.

## 4. Discussion

Our results showed that performing antenatal aquatic aerobic exercises of moderate intensity reduced PPD symptoms and increased the quality of life of mothers. Our findings partially supported our initial hypothesis; the percentage of women identified as having a risk of depression (EPDS > 10) was also lower in the intervention group, but this difference did not reach statistical significance.

Pregnancy and delivery bring many physiological and psychosocial changes that can alter a woman’s ability to perform her usual roles and affect her perceived quality of life. Existing literature suggest that physical activity during pregnancy improve women’s subjective perception of their health-related quality of life [49], mainly in the physical health-related quality of life [49,62,63]. In our study, we found a reduction of the EQ-5D domains and this reduction were statistically significant in the anxiety and depression dimension.

Fatigue and sleep problems during pregnancy are a result of hormonal and mechanical changes [64]. Previous studies have shown that sleep problems during pregnancy are associated with an increased risk of depression both during pregnancy and postpartum [65,66]. Poor sleep quality is related to a worse quality of life related to health on all dimensions except the emotional role [44]. We found a reduction in the global Sleep Problems Index and almost all specific sleep problems evaluated by the MOS sleep questionnaire (except for awakening with short breath or headache) in the exercise group compared to the control group, although these differences were not statistically significant.

Previous studies showed that postpartum exercise decreased PPD [32,35], but the effect of exercise during pregnancy on subsequent PPD is controversial. Our results suggest a promising benefit of antenatal exercise on PPD, in that it reduced some of the symptoms, consistent with previous studies [37,38]. Vargas-Terrones et al. [37] performed a small study in which women engaged in moderate exercise for 60 min three days per week and found a 15% reduction in PPD. Aguilar-Cordero et al. [38] performed a study using an intervention similar to ours (antenatal aquatic exercise) and found that their intervention reduced PPD symptoms. Performing aerobic exercises in water is a low-impact form of exercise that is less harmful than exercise on land. Aquatic exercise has positive physiologic effects generated by thermal dissipation that may improve blood circulation and venous return, and alleviate pregnancy-induced edema [67]. Furthermore, aerobic exercise can be rewarding because it involves all major muscles and includes stretching, breathing, and relaxation techniques. Exercising with a group also provides socialization and emotional interactions among women. Either or both of these could explain the beneficial effects of water aerobic exercise on PPD.

In contrast, three randomized controlled trials (RCTs) that studied the effects of exercise during pregnancy on subsequent PPD found no effect [39,40,68]. Two of these RCTs [39,40] examined the impact of supervised physical exercise with a different number of sessions per week; Songoygard et al. [40] examined the effect of one 1 h session per week and Coll et al. [39] examined the effect of three 1 h sessions per week. However, Mohammadi et al. [68] simply examined the effect of providing advice for performing three 20–30 min sessions per week.

The different effects of exercise during pregnancy on PPD reported by these different studies could be explained by differences in the duration, frequency, and intensity of the exercise, and whether the exercise was supervised or was only a recommendation. A systematic review concluded that to achieve a reduction in symptoms of depression, pregnant women needed to accumulate at least 644 MET-min/week [34]. Our intervention group performed exercise 3 days per week at moderate intensity (150 min per week) and reached the recommended exercise intensity. Some studies reported that depression and anxiety during pregnancy are two of the most important risk factors for developing PPD [69,70]. The variations in the benefit provided by antenatal exercise among different studies may also be because PPD was evaluated at different times after childbirth (1 month (present study), 6 weeks [37,38], 8 weeks [68], and 12 weeks [39,40]).

A substantial body of evidence has reported that exercise during pregnancy can prevent weight gain [30] and gestational diabetes [71] and reduce the risk of hypertension [72]. Furthermore, exercise during pregnancy is safe for women who have no contraindications for exercise, and the WHO recently recommended at least 150 min per week of moderate-intensity aerobic physical activity for pregnant women [73]. Importantly, our intervention, a moderate-intensity aerobic water exercise, was safe for the mother and the newborn [74]. Importantly, because water aerobics is a low-impact exercise, it is safer than weight-bearing exercise on land [75,76].

Although exercise during pregnancy has beneficial effects, the optimal frequency, intensity, and exercise modality remain to be determined.

### Study Strengths and Limitations

The strengths of this study are that it was a large and multi-site RCT, and the intervention was delivered by midwives who provided integral care during pregnancy and childbirth, data were analyzed using ITT analyses, and outcomes were evaluated using blinded analyses. Although the exercise and usual practice groups were homogenous and well balanced in most baseline characteristics, they differed in smoking status, social class, and baseline physical activity. We therefore performed a sensitivity analysis by adjusting for baseline social class and smoking status to ascertain their effects. The results indicated no relevant differences in the magnitude or statistical significance of the outcomes (data not shown).

Nonetheless, our results should be interpreted with caution and require confirmation by subsequent research because they are derived from secondary study objectives. Although we evaluated the effects of exercise during pregnancy on the prevention of PPD in a healthy population of pregnant women, we did not assess depressive symptoms at baseline, and some of the participants may have been depressed during pregnancy. We did not quantify any potential physiological mediating variable; therefore, it is possible that the results of our study also reflect the effect of social interaction during pregnancy. Another limitation is that we measured PPD once during the postpartum period, and have no information on PPD during later months (e.g., 12 weeks post-delivery); thus, we should not discard an effect attenuation at longer term.

## 5. Conclusions

Aquatic aerobic exercise during pregnancy decreased postpartum anxiety and depressive symptoms in mothers, and had no adverse effects on the mother or the newborn. Aquatic exercise during pregnancy should therefore be considered for the prevention of PPD.

## Figures and Tables

**Figure 1 jcm-10-02432-f001:**
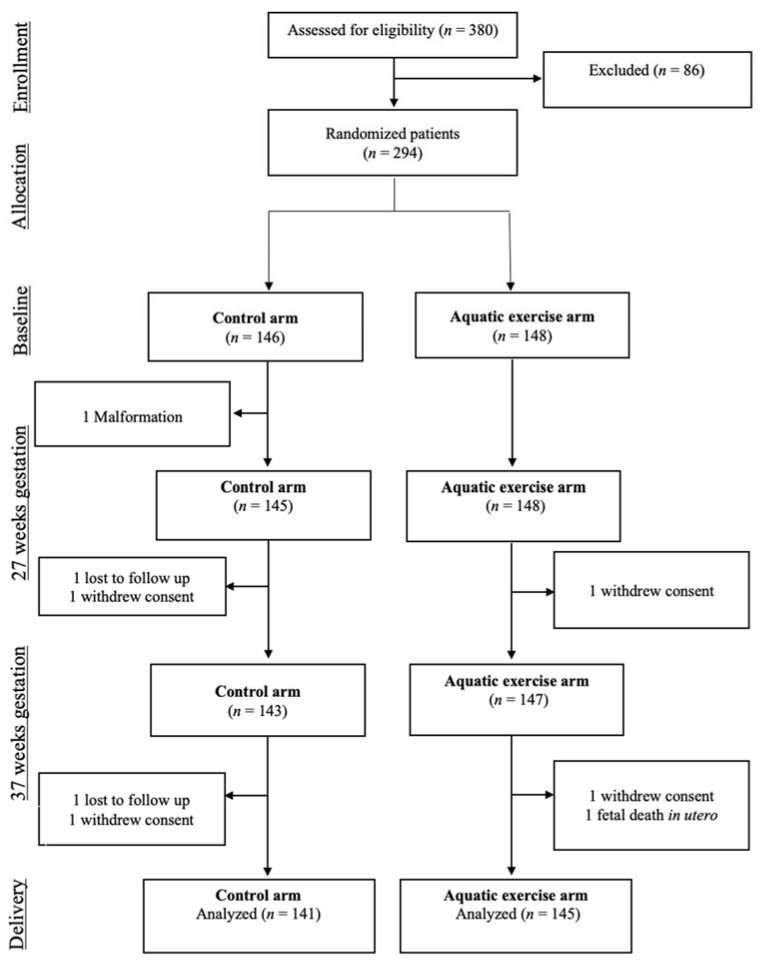
Disposition of study participants.

**Table 1 jcm-10-02432-t001:** Baseline characteristics of patients in the aquatic exercise (intervention) group and the control (usual care) group.

	Group	
	Intervention (*n* = 148)	Control (*n* = 146)
Age (years), mean ± SD	31.1 ± 4.1	31.5 ± 4.2
BMI (kg/m^2^), mean ± SD	23.5 ± 3.2	23.4 ± 3.1
Obstetric characteristics, n/N (%)		
Nulligravid	98/145 (67.6)	98/141 (69.5)
Preterm	1/145 (0.7)	1/141 (0.7)
Spontaneous abortion	39/145 (26.9)	39/141 (27.6)
One previous child	45/145 (21.0)	38/141 (27.0)
Two or more previous children	3/145 (2.1)	4/141 (2.8)
Smoking, n/N (%)		
Smoker at inclusion	11/143 (17.9)	25/140 (7.7)
Quit smoking prior to pregnancy	12/143 (8.4)	24/140 (17.1)
Social class, n/N (%)		
I and II	49/124 (39.5)	47/126 (37.3)
III	42/124 (33.9)	33/126 (26.2)
IV and V	33/124 (26.6)	46/126 (36.5)
Educational level, n/N (%)		
Primary school	8/147 (5.4)	14/143 (9.8)
Secondary school	61/147 (41.5)	54/143 (37.8)
University	78/147 (51.7)	74/143 (51.7)
IPAQ (MET-min/week), mean ± SD	1008.0 ± 14,033.3	1047 ± 10,33.3
IPAQ category score, n/N (%)		
Low	63/145 (43.4)	75/141 (53.2)
Moderate	76/145 (52.4)	60/141 (42.6)
High	6/145 (4.1)	6/141 (4.3)
MOS sleep, mean ± SD		
Sleep disturbance	29.9 ± 22.6	25.9 ± 20.6
Snoring	12.6 ± 22.6	16.4 ± 28.2
Awakening with short breath or headache	12.0 ± 19.2	13.6 ± 22.2
Sleep adequacy	60.6 ± 28.1	60.9 ± 29.2
Day-time somnolence	37.4 ± 16.9	37.2 ± 19.2
Sleep Problems Index	20.8 ± 13.7	21.6 ± 14.4
MOS sleep: optimal sleep	93/147 (63.3)	91/145 (62.8)
EQ5D dimension		
Mobility problems	7/147 (4.8)	7/144 (4.9)
Self-care problems	1/147 (0.7)	2/144 (1.4)
Usual activity problems	7/146 (4.8)	11/142 (7.7)
Pain/discomfort problems	46/147 (31.3)	55/144 (38.2)
Anxiety/depression	20/147 (13.6)	15/144 (10.4)
EQ VAS, mean ± SD	86.4 ± 77.9	89.6 ± 76.8

SD: standard deviation; BMI: body mass index; IPAQ: International Physical Activity Questionnaire; MET: Metabolic Equivalent of Task; MOS: Medical Outcome Study; EQ5D: EuroQol 5-Dimension; EQ VAS: EuroQol Visual Analogue Scale.

**Table 2 jcm-10-02432-t002:** Outcomes of the intervention group and control group.

	Intervention	Control	OR (95% CI)	*p*	ITT Analysis: Imputed OR (95% CI)	*p*
**EQ5D dimension, *n/N* (%)**						
Mobility problems	5/139 (3.6)	7/132 (5.3)	0.66 (0.20, 2.14)	0.491	0.66 (0.20, 2.14)	0.497
Self-care problems	2/139 (1.4)	0/132 (0)	NA	-	NA	-
Usual activity problems	4/137 (2.9)	9/130 (6.9)	0.41 (0.12, 1.38)	0.151	0.41 (0.12, 1.38)	0.158
Pain/discomfort problems	21/139 (15.1)	31/132 (23.5)	0.59 (0.32, 1.10)	0.102	0.59 (0.32, 1.10)	0.11
Anxiety/depression	16/139 (11.5)	30/132 (22.7)	0.42 (0.21, 0.82)	0.011	0.42 (0.21, 0.82)	0.02
			**Beta (95% CI)**		**Imputed Beta**	
					**(95% CI)**	
**EQ VAS, mean ± SD**	86.8 (14.4)	82.5 (13.7)	−19.45 (−61.72, 22.83)	0.366	−19.44 (−61.28, 22.41)	0.361
**Depression Edinburgh questionnaire total (EDPS)**	6.1 ± 1.9	6.8 ± 2.4	−0.68 (−1.20, −0.15)	0.012	−0.70 (−1.24, −0.17)	0.01
			**OR (95% CI)**	**p**	**ITT analysis: Imputed OR (95% CI)**	***p***
**Risk Depression Edinburgh questionnaire**	2/136 (1.5)	8/132 (6.1)	0.231 (0.048, 1.11)	0.067	0.211 (0.044, 1.01)	0.052
			**Beta (95% CI)**		**Imputed Beta**	***p***
					**(95% CI)**	
**MOS-Sleep**						
Sleep disturbance	38.4 ± 22.9	40.9 ± 24.8	−3.99 (−9.47, 1.58)	0.161	−3.95 (−9.46,1.55)	0.159
Snoring	26.5 ± 33.8	36.1 ± 37.5	−5.78 (−13.03, 1.48)	0.118	−5.95 (−13.2, 1.31)	0.108
Awakening with short breath	7.2 ± 16.1	7.1 ± 14.3	0.25 (−3.35, 3.86)	0.89	0.26 (−3.3, 3.87)	0.886
or headache						
Sleep adequacy	56.0 ± 28.5	56.1 ± 30.1	−0.87 (−7.43, 5.68)	0.793	−0.88 (−7.46, 5.68)	0.791
Day-time somnolence	32.0 ± 16.2	34.4 ± 18.3	−1.80 (−5.77, 2.18)	0.375	−1.69 (−5.67, 2.27)	0.4
Sleep Problems Index	32.0 ± 18.6	31.2 ± 18.0	−0.74 (−4.71, 3.22)	0.713	−0.65 (−4.58, 3.28)	0.746
			**OR (95% CI)**	**p**	**ITT analysis: Imputed OR (95% CI)**	***p***
MOS-sleep optimal sleep	64/130 (49.2)	74/129 (57.4)	0.69 (0.42, 1.14)	0.148	0.69 (0.42, 1.14)	0.152

SD: standard deviation; MOS: Medical Outcome Study; EQ5D: EuroQol 5-Dimension; EQ VAS: EuroQol Visual Analogue Scale; ITT: intention to treat.

**Table 3 jcm-10-02432-t003:** Maternal adverse events and newborn outcomes in the intervention and control groups.

Mother (Adverse Event), *n/N* (%)	Intervention	Control	*p*
Urinary tract infection	7/139 (5.0)	11/132 (8.3)	0.276
Back pain	3/139 (2.2)	1/132 (0)	0.623
Bleeding	3/139 (2.2)	2/132 (1.5)	1
Contractions	3/139 (2.2)	0/132 (0)	0.248
Abortion/fetal death	1/139 (0.7)	1/132 (0.8)	1
Fetal admission to intensive care	10/139 (7.2)	5/132 (3.8)	0.288
Mother admission to intensive care	1/139 (0.7)	0/132 (0)	1
**Newborn, *n/N* (%)**			
Weight, g ± SD	3367 ± 799.7	3281 ± 497.1	0.283
Intrapartum fetal distress, n (%)	25/120 (20.8)	28/119 (23.5)	0.365
Weeks of gestation	39.9 ± 2.0	39.8 ± 2.0	0.739
At least 41 weeks gestation	20/139 (14.4)	17/132 (12.9)	0.725
Apgar score			
1 min	8.7 ± 1.3	8.7 ± 0.9	0.813
5 min	9.8 ± 0.99	9.8 ± 0.4	0.398
pH of umbilical cord blood	7.27 ± 0.09	7.26 ± 0.07	0.42

## Data Availability

Data presented in this study are available on request from the corresponding author. An anonymized minimal dataset is available (https://doi.org/10.5281/zenodo.3580637).

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
