# Peer review of "Effectiveness of Moderate-Intensity Aerobic Water Exercise during Pregnancy on Quality of Life and Postpartum Depression: A Multi-Center, Randomized Controlled Trial"

_jcm, 2021, doi:10.3390/jcm10112432_

Round 1
Reviewer 1 Report
This manuscript presents the results of an RCT testing the effectiveness of a physical exercise intervention during pregnancy in improving postnatal quality of life and preventing postpartum depression. Outcomes showed that women in the intervention group had fewer symptoms of postpartum depression. While the paper discusses a relatively novel topic, I have some comments and suggestions, which I hope the authors can address; these should considerably strengthen this contribution.
While the introduction is very comprehensive in describing the epidemiology of depression and postpartum depression, I would have liked to see some more background related to the specific focus of this paper, i.e. (antenatal) exercise and postpartum depression. In particular, the authors note contradictory findings reported in previous studies (p2, l79-81). It would have been relevant to go into more detail of what these contradictions pertained to (is it the effects of exercise on postpartum depression in general ; or more specifically the type, duration, intensity of the exercise?). Finally, given the overall objective of the study, some more description on the role of exercise relative to sleep problems and quality of life in postpartum women would have been appropriate.
Methods
Paragraph 2.4: While the study protocol is available elsewhere, a short repetition of the rationale why an aquatic intervention was chosen would have benefited the reader. When reading the protocol, this is particularly important as the primary aim of the intervention was reducing use of epidural analgesia during labor, for which other intervention content choices might apply than for the prevention of PPD.
Paragraph 2.6: l158-160 – did the effectiveness analyses also account for the differences in physical activity, stated on p5, l189? In the context of intervention research, intervention adherence and execution would also seem to be an important confounder that should be explored in sensitivity analyses.
L162-163: please detail through which methods MI was achieved
Results
It is not clearly described how the different outcomes were tested, but if for instance each subscale for the EQ-5D and MOS-sleep was tested separately for potential effects, there is a serious risk for multiple testing effects.
Discussion
Like the introduction, the discussion primarily focuses on the role of exercise on PPD and neither on quality of life nor sleep. The lack of effects on these outcomes should also be discussed and, where possible, compared to previous literature on this topic.
For the control group, was there any information on their physical activity patterns during pregnancy, including swimming? Potential differences in exercise patterns during pregnancy should also be considered in the discussion section and related to the presented intervention effects.
Postpartum depression was measures at 1 month after birth. However, most PPD present themselves around 6 weeks to two months after birth. Thus it might be possible that some cases were not detected, inflating the overall effects of the intervention on this outcome. Although the authors briefly touch upon this subject, this is one of the major limitations of the study and warrants more discussion.
Conclusions do not completely fit the reported results: the overall outcomes for quality of life were not significant and the only subscale that showed preference for the intervention group was the Anxiety/depression subscale, a result pointing in the same direction as the analyses with the EPDS.
Author Response
Dear Editor and reviewers,
We would like to thank the Editor and reviewers the time dedicated to revising the manuscript, their assessments and all the comments made on this work. We agree with most of the observations and are convinced that have improved the clarity and scientific value of this research. We apologize for the mistakes, and we have also corrected the text as suggested. Track changes from Microsoft Word show the modifications performed by the authors. To make the changes easier to understand, we have including, when possible, the revised added texts in the manuscript as answers to your comments.
Thank you very much for your time and interest.
We look forward to hearing from you soon.
Sincerely,
Miquel Bennasar-Veny
Reviewer 1
This manuscript presents the results of an RCT testing the effectiveness of a physical exercise intervention during pregnancy in improving postnatal quality of life and preventing postpartum depression. Outcomes showed that women in the intervention group had fewer symptoms of postpartum depression. While the paper discusses a relatively novel topic, I have some comments and suggestions, which I hope the authors can address; these should considerably strengthen this contribution.
While the introduction is very comprehensive in describing the epidemiology of depression and postpartum depression, I would have liked to see some more background related to the specific focus of this paper, i.e. (antenatal) exercise and postpartum depression. In particular, the authors note contradictory findings reported in previous studies (p2, l79-81). It would have been relevant to go into more detail of what these contradictions pertained to (is it the effects of exercise on postpartum depression in general; or more specifically the type, duration, intensity of the exercise?).
R: We thank the reviewer for the insightful comment. We have included more background about exercise during pregnancy and postpartum depression. Also, we have clarified the contradictory findings in previous studies (page 2, lines 85-88): “These contradictory results could be explained by the type, intensity, frequency and duration of the physical activity. Also, it could be due to the supervision of the exercise or the combination of physical activity the with other interventions (diets, education, smoking) [37-40]”.
Finally, given the overall objective of the study, some more description on the role of exercise relative to sleep problems and quality of life in postpartum women would have been appropriate.
R: Thank you for pointing this out. We have included in the introduction section (page 2, lines 89-96): “Women in the postpartum period experience high levels of sleep disturbances [41,42]. Previous studies have shown that sleep problems adversely affect mood [43] and reduce quality of life in mothers [44]. Sleep deprivation is also associated with antenatal stress and PPD [45,46]. Women in the postpartum period experience high levels of sleep disturbances due to irregular newborn sleep patterns and the demands of the newborn [41,42]. The association between sleep and depression may result from similar mechanisms [47]. Some studies have shown that exercise, especially water exercise, can improve the sleep quality of pregnant women [48,49]”.
Methods
Paragraph 2.4: While the study protocol is available elsewhere, a short repetition of the rationale why an aquatic intervention was chosen would have benefited the reader. When reading the protocol, this is particularly important as the primary aim of the intervention was reducing use of epidural analgesia during labor, for which other intervention content choices might apply than for the prevention of PPD.
R: Thank you very much for your suggestion. The main reason to choose aquatic intervention was related with the low impact of this type of exercise but also taking into account emotional benefits of group interaction. We have now added the rationale for aquatic intervention (page 2, lines 74-76): “Aquatic-aerobic exercise has some advantages including lower probability of miscarriage, less edema, increased diuresis, reduced arterial pressure and less back pain [26] and also allow for emotional interactions among pregnant women [27]”.
Paragraph 2.6: l158-160 – did the effectiveness analyses also account for the differences in physical activity, stated on p5, l189? In the context of intervention research, intervention adherence and execution would also seem to be an important confounder that should be explored in sensitivity analyses.
R: We performed a sensitivity analysis adjusted by baseline physical activity (MET-min/week). The findings are consistent with those from the analysis. P-values and magnitudes of the differences in intervention and control group did not change after adjusting for baseline physical activity. Differences in Depression Edinburgh questionnaire, and anxiety/depression dimension of the EuroQol Questionnaire were statistically significance after adjustment, with no differences in p-values. We included in the manuscript the description of the sensitivity analysis performance.
We added in the Statistical analysis section: “Sensitivity analysis included the effectiveness of the intervention adjusted for unbalanced variables at baseline” (page 4, lines 188-190) and in the Study strengths and limitations section: “Although the exercise and usual practice groups were homogenous and well balanced in most baseline characteristics, they differed in smoking status, social class and baseline physical activity” (page 10, lines 323-325).
L162-163: please detail through which methods MI was achieved
R: Thank you for pointing this out. We agree with this comment. We have included in the methods section (page 4, lines 191-194): “Multiple imputation was used for the main analysis because it generally provides less biased estimates of an effect than a complete cases analysis. Missing outcomes were accounted for using multiple imputation with chained equation [60]”.
Results
It is not clearly described how the different outcomes were tested, but if for instance each subscale for the EQ-5D and MOS-sleep was tested separately for potential effects, there is a serious risk for multiple testing effects.
R: We tested the five dimension and the EQ-VAS of EQ-5D, and the six dimension and the total sleep problem index of the MOS-sleep as indicated by their authors. The case of multiplicity in clinical trial should be treated differently from other epidemiological designs, as stated in Li et al. (2017): Clinical trials often assess multiple outcomes (or ‘endpoints’) such as symptoms, blood test results, side effects, quality of life, or death, to try to maximize the usefulness of information from a costly trial. For example, in a cardiovascular trial, outcomes of interest may include hospitalization, stroke, heart failure, myocardial infarction, cardiac arrest, disability and death….
To avoid inflation of the type I error rate, several solutions have been proposed. The first option is to identify one single outcome as the primary outcome and to treat the remaining outcomes as secondary in the study design. There is no need to adjust for multiplicity when there is a single primary outcome, as findings for secondary outcomes are considered subsidiary and exploratory, rather than confirmatory.
(Li G, Taljaard M, Van den Heuvel ER, Levine MA, Cook DJ, Wells GA, Devereaux PJ, Thabane L. An introduction to multiplicity issues in clinical trials: the what, why, when and how. Int J Epidemiol. 2017 Apr 1;46(2):746-755).
We indicated in the strength and limitation section that finding should be carefully interpreted: “Nonetheless, our results should be interpreted with caution and require confirmation by subsequent research because they are derived from secondary study objectives” (page 10, lines 330-331).
Discussion
Like the introduction, the discussion primarily focuses on the role of exercise on PPD and neither on quality of life nor sleep. The lack of effects on these outcomes should also be discussed and, where possible, compared to previous literature on this topic.
R: Thank you for pointing this out. We have included in the discussion section (page 8-9, lines 260-275): “Pregnancy and delivery bring many physiological and psychosocial changes that can alter a woman's ability to perform her usual roles and affect her perceived quality of life. Existing literature suggest that physical activity during pregnancy improve women's subjective perception of their health-related quality of life [49], mainly in the physical health-related quality of life [49,62,63]. In our study, we found a reduction of the EQ-5D domains and this reduction were statistically significant in the anxiety and depression dimension.
Fatigue and sleep problems during pregnancy are a result of hormonal and mechanical changes [64]. Previous studies have shown that sleep problems during pregnancy are associated with an increased risk of depression both during pregnancy and postpartum [65,66]. Poor sleep quality is related to a worse quality of life related to health on all dimensions except the emotional role [44]. We found a reduction in on the global sleep problems index and almost all specific sleep problems evaluated by the MOS sleep questionnaire (except for awakening with short breath or headache) in the exercise group compared to the control group, although these differences were not statistically significant”.
For the control group, was there any information on their physical activity patterns during pregnancy, including swimming? Potential differences in exercise patterns during pregnancy should also be considered in the discussion section and related to the presented intervention effects.
R: We increased the sample size a 15% assuming that the effect size of the intervention would be small if the control group also exercise (Navas et al. Effectiveness and safety of moderate-intensity aerobic water exercise during pregnancy for reducing use of epidural analgesia during labor: protocol for a randomized clinical trial. BMC Pregnancy Childbirth 2018, 18, 94, doi:10.1186/s12884-018-1715-3).
We assumed that women in the control group would also engage in physical activity, we hypothesized that women in the intervention group would exercise more, but we did not collect physical activity in the control group.
Postpartum depression was measures at 1 month after birth. However, most PPD present themselves around 6 weeks to two months after birth. Thus it might be possible that some cases were not detected, inflating the overall effects of the intervention on this outcome. Although the authors briefly touch upon this subject, this is one of the major limitations of the study and warrants more discussion.
R: Thank you for your suggestion. We agree with the reviewer that some PPD could not be detected at 1 month, but probably most cases manifest early symptoms of depression that could be detected with the EDPS. In any case, this possibility would occur in both groups that we are comparing (intervention and control) thus the overall effects will not be biased but could be attenuated during follow up. We have added this consideration in the discussion section (page 10, lines 336-339): “Another limitation is that we measured PPD once during the postpartum period and have no information on PPD during later months (e.g., 12 weeks postdelivery), thus we should not discard an effect attenuation at longer term”.
Conclusions do not completely fit the reported results: the overall outcomes for quality of life were not significant and the only subscale that showed preference for the intervention group was the Anxiety/depression subscale, a result pointing in the same direction as the analyses with the EPDS.
R: We thank the reviewer for the insightful comment, we have changed the conclusion section (page 10, lines 341-342): “Aquatic aerobic exercise during pregnancy decreased postpartum anxiety and depressive symptoms in mothers and had no adverse effects on the mother or the newborn”.

Reviewer 2 Report
General Comments:
The present study aimed to analyze the effectiveness and safety of a moderate aerobic water exercise program on the risk of postpartum depression, sleep problems, and quality of life in women at 1 month after delivery. Overall, the results showed that women in the intervention group were less likely to report anxiety or depression and had a lower mean EPDS score. The authors have presented a well-written manuscript with coherence between sections. The selection of references to support the study is updated. The present study is original, the rationale is well described, the research purpose is relevant, and the statistical analyses are well-designed. The investigators did not present a study hypothesis and did not provide the observed power and/or effect size of the analyses. There are some points that need to be addressed before the manuscript may be considered for publication in order to improve the comprehension of the information. Please refer to the “Specific Comments” below. To the investigators, I wish you and your families health and some semblance of stability during these pandemic times.
Specific Comments:
Introduction:
- Please add a priori hypothesis for the study.
Participants:
- Line 93: Please remove the word ‘fetus’.
- Line 94: Please add a definition and/or a reference to support ‘low obstetric risk’.
Intervention:
- Please provide more information about who delivered the intervention.
Results:
- Please consider adding a description of the sample size calculation.
- Please consider providing the observed power and/or effect size of the analyses.
- Flowchart: Please fix the ‘n’ of the control arm at 37 weeks; 143 instead of 133.
Discussion:
- Line 231-232: Please consider re-write the sentence “Performing aerobic exercises in water can be rewarding, because it involves all major muscles and includes stretching, breathing, and relaxation techniques.” by adding a more physiologic point-of-view of the differences between water and land exercises.
- Line 259-260: Please consider also adding the reference https://doi.org/10.1016/j.clinbiomech.2019.05.021 to support the safety of water-aerobics during pregnancy.
- Please consider findings from the study https://doi.org/10.1136/bjsports-2018-099697 for your discussion.
- Line 277-278: Please consider removing the sentence “However, there is no reason to believe the groups differed in this characteristic due to the similarity of baseline characteristics.”
- Please add a comment if the findings were aligned or not with the initial hypothesis of the study.
- Please provide a description on how the research team proceeded when receiving a questionnaire EDPS > 10. Did the research team provide any support for those participants?
Conclusion:
- Line 283-284: Please consider re-writing the sentence “Aquatic exercise during pregnancy decreased the symptoms of subsequent PPD, improved the physical and mental health of women, increased their quality of life,…” because it´s not possible to know if the intervention actually decreased the symptoms of PPD, as no baseline data were collected to allow pre-post comparisons; also, the only significant difference in EQ5D was the anxiety/depression dimension, thus investigators should make conclusions accordingly. The conclusion described in the abstract seems more adequate.
Data Availability Statement:
- Line 303-304: Please consider adding the link (https://doi.org/10.5281/zenodo.3580637) for the minimal dataset.
Table 2:
- Please consider adding notes to table 2 similarly to what was added to table 1.
References:
- Please consider deleting #26 as the updated version (#24) is also provided.
Author Response
Dear Editor and reviewers,
We would like to thank the Editor and reviewers the time dedicated to revising the manuscript, their assessments and all the comments made on this work. We agree with most of the observations and are convinced that have improved the clarity and scientific value of this research. We apologize for the mistakes, and we have also corrected the text as suggested. Track changes from Microsoft Word show the modifications performed by the authors. To make the changes easier to understand, we have including, when possible, the revised added texts in the manuscript as answers to your comments.
Thank you very much for your time and interest.
We look forward to hearing from you soon.
Sincerely,
Miquel Bennasar-Veny
Reviewer 2
General Comments:
The present study aimed to analyze the effectiveness and safety of a moderate aerobic water exercise program on the risk of postpartum depression, sleep problems, and quality of life in women at 1 month after delivery. Overall, the results showed that women in the intervention group were less likely to report anxiety or depression and had a lower mean EPDS score. The authors have presented a well-written manuscript with coherence between sections. The selection of references to support the study is updated. The present study is original, the rationale is well described, the research purpose is relevant, and the statistical analyses are well-designed. The investigators did not present a study hypothesis and did not provide the observed power and/or effect size of the analyses. There are some points that need to be addressed before the manuscript may be considered for publication in order to improve the comprehension of the information. Please refer to the “Specific Comments” below. To the investigators, I wish you and your families health and some semblance of stability during these pandemic times.
R: We would like to thank the reviewer for careful and thorough reading of this manuscript.
Specific Comments:
Introduction:
- Please add a priori hypothesis for the study.
R: We have now added the a priori hypothesis for the study (page 3, lines 99-100): “We hypothesized that women in moderate aerobic water exercise are at less risk of postnatal depression than women in the control group”.
Participants:
- Line 93: Please remove the word ‘fetus’.
R: Thanks, we have amended (page 3, line 109).
- Line 94: Please add a definition and/or a reference to support ‘low obstetric risk’.
R: Thank you, we have included a definition and a reference to support “low obstetric risk” (page 3, lines 110-112): “low-risk women with uncomplicated singleton term pregnancies with a vertex presentation who are expected to have an uncomplicated birth [51]”.
Intervention:
- Please provide more information about who delivered the intervention.
R: Thank you for pointing this out. We agree with this comment. We have included in the intervention section (page 3, line 144): “The aquatic aerobic exercise program was carried out exclusively by midwives and consisted…”
Results:
- Please consider adding a description of the sample size calculation.
- Please consider providing the observed power and/or effect size of the analyses.
R: This information has been added (page 3, lines 122-127):
“2.3. Sample size
For the primary hypothesis 320 women provided 80% power at 5% significance two tailed alpha to detect differences in the primary outcome. For secondary hypothesis, we had 80% power to detect a between-arm effect size of 0.22 (standardized difference) in Depression Edinburgh questionnaire and MOS-sleep questionnaire and a reduction of 5% in the risk of post-partum depression”.
- Flowchart: Please fix the ‘n’ of the control arm at 37 weeks; 143 instead of 133.
R: Sorry for this mistake. We have amended.
Discussion:
- Line 231-232: Please consider re-write the sentence “Performing aerobic exercises in water can be rewarding, because it involves all major muscles and includes stretching, breathing, and relaxation techniques.” by adding a more physiologic point-of-view of the differences between water and land exercises.
R: Thank you for pointing this out. We agree with this comment and we have changed the sentence (page 9, lines 283-290): “Performing aerobic exercises in water is a low-impact form of exercise that is less harmful than exercise on land. Aquatic exercise has physiologic positive effects generated by thermal dissipation that may improve blood circulation and venous return, alleviate pregnancy-induced edema [67]. Furthermore, aerobic exercise can be rewarding, because it involves all major muscles and includes stretching, breathing, and relaxation techniques. Exercising with a group also provides socialization and emotional interactions among women. Either or both of these could explain the beneficial effects of water aerobic exercise on PPD”.
- Line 259-260: Please consider also adding the reference https://doi.org/10.1016/j.clinbiomech.2019.05.021 to support the safety of water-aerobics during pregnancy.
R: Thank you for your suggestion. We have included (page 9, line 315).
- Please consider findings from the study https://doi.org/10.1136/bjsports-2018-099697 for your discussion.
R: We did not include the systematic review and meta-analysis of Davenport et. al in the discussion (but it is included in the introduction) because this meta-analysis did not include relevant evidence of RCT on postnatal depression from 3 RCTs published in 2019. The systematic review of Davenport included only 5 RCTs until 2018 of exercise alone and some were small and very low-quality clinical trials. Davenport et al also include unsupervised exercised in combination with other interventions. We finally decided to not include systematic reviews in the discussion because of the heterogeneity of the RCTs but discuss the RCTs.
- Line 277-278: Please consider removing the sentence “However, there is no reason to believe the groups differed in this characteristic due to the similarity of baseline characteristics.”
R: We have removed the sentence as the reviewer suggest.
- Please add a comment if the findings were aligned or not with the initial hypothesis of the study.
R: Following your suggestion we have added “Our findings partially supported our initial hypothesis” (page 8; lines 256-257).
- Please provide a description on how the research team proceeded when receiving a questionnaire EDPS > 10. Did the research team provide any support for those participants?
R: Thank you for pointing this out. We have included in the XXXXX section (page 5, lines 204-205): “Women at risk of depression (EPDS>10) were referred to primary health care service by the research team”.
Conclusion:
- Line 283-284: Please consider re-writing the sentence “Aquatic exercise during pregnancy decreased the symptoms of subsequent PPD, improved the physical and mental health of women, increased their quality of life,…” because it´s not possible to know if the intervention actually decreased the symptoms of PPD, as no baseline data were collected to allow pre-post comparisons; also, the only significant difference in EQ5D was the anxiety/depression dimension, thus investigators should make conclusions accordingly. The conclusion described in the abstract seems more adequate.
R: We thank the reviewer for the suggestion, we have changed the conclusion of the study (page 10, lines 341-342): “Aquatic aerobic exercise during pregnancy decreased postpartum anxiety and depressive symptoms in mothers and had no adverse effects on the mother or the newborn”.
Data Availability Statement:
- Line 303-304: Please consider adding the link (https://doi.org/10.5281/zenodo.3580637) for the minimal dataset.
R: Thanks for this comment, we have included the link for the minimal dataset in the Data Availability Statement (page 10, lines 362-363).
Table 2:
- Please consider adding notes to table 2 similarly to what was added to table 1.
R: Thank you, we have amended it.
References:
- Please consider deleting #26 as the updated version (#24) is also provided.
R: Thank you for this suggestion, we have changed it following your instructions.

Reviewer 3 Report
Since the exercise did not include any physiological quantification, the results most likely reflect a response to a social interaction event.
Inclusion of an additional study group practicing other relaxation techniques, and compare to swimming ,could have been interesting.
Author Response
Dear Editor and reviewers,
We would like to thank the Editor and reviewers the time dedicated to revising the manuscript, their assessments and all the comments made on this work. We agree with most of the observations and are convinced that have improved the clarity and scientific value of this research. We apologize for the mistakes, and we have also corrected the text as suggested. Track changes from Microsoft Word show the modifications performed by the authors. To make the changes easier to understand, we have including, when possible, the revised added texts in the manuscript as answers to your comments.
Thank you very much for your time and interest.
We look forward to hearing from you soon.
Sincerely,
Miquel Bennasar-Veny
Reviewer 3
Since the exercise did not include any physiological quantification, the results most likely reflect a response to a social interaction event.
Inclusion of an additional study group practicing other relaxation techniques, and compare to swimming, could have been interesting.
R: We agree with the reviewer and we have added this consideration in the discussion section (page 10, lines 334-336): “We did not quantify any potential physiological mediating variable, therefore it is possible that the results of our study also reflect the effect of social interaction during pregnancy”.

Round 2
Reviewer 1 Report
I am satisfied by the way the authors have included the reviewers' comments in the adapted version of their manuscript